# Protein-Based High Internal Phase Pickering Emulsions: A Review of Their Fabrication, Composition and Future Perspectives in the Food Industry

**DOI:** 10.3390/foods12030482

**Published:** 2023-01-20

**Authors:** Minghao Zhang, Xiang Li, Li Zhou, Weilin Chen, Eric Marchioni

**Affiliations:** 1National Demonstration Center for Experimental Ethnopharmacology Education, School of Pharmaceutical Sciences, South-Central MinZu University, Wuhan 430074, China; 2Inst Pluridisciplinaire Hubert Curien, CNRS, Equipe Chim Analyt Mol Bioact & Pharmacognoise, UMR 7178, UDS, F-67400 Illkirch Graffenstaden, France

**Keywords:** protein, high internal phase, Pickering emulsions, fabrication, composition

## Abstract

Protein-based high internal phase Pickering emulsions (HIPEs) are emulsions using protein particles as a stabilizer in which the volume fraction of the dispersed phase exceeds 74%. Stabilizers are irreversibly adsorbed at the interface of the oil phase and water phase to maintain the droplet structure. Protein-based HIPEs have shown great potential for a variety of fields, including foods, due to the wide range of materials, simple preparation, and good biocompatibility. This review introduces the preparation routes of protein-based HIPEs and summarizes and classifies the preparation methods of protein stabilizers according to their formation mechanism. Further outlined are the types and properties of protein stabilizers used in the present studies, the composition of the oil phase, the encapsulating substances, and the properties of the constituted protein-based HIPEs. Finally, future development of protein-based HIPEs was explored, such as the development of protein-based stabilizers, the improvement of emulsification technology, and the quality control of stabilizers and protein-based HIPEs.

## 1. Introduction

An emulsion consists of two immiscible liquids (usually oil and water), with one liquid dispersed as small droplets in the other, known as the dispersed (internal) phase and the continuous (external) phase, respectively. Based on the spatial distribution of the liquids, emulsions can be roughly divided into oil-in-water emulsions (O/W), water-in-oil emulsions (W/O), or more complex multiple emulsions (O_1_/W/O_2_, or W_1_/O/W_2_) [1]. When emulsion droplets collide with neighboring droplets, they tend to merge, which is a thermodynamically unstable system [2]. In 1907, Pickering pointed out that oil droplets in emulsions wrapped in a film of solid particles smaller than oil droplets could prevent them from destabilizing [3]. These emulsions stabilized by solid particles wetted by both liquids are defined as Pickering emulsions. Currently, the Pickering emulsion stabilization mechanism has been investigated intensively, deriving from the interfacial behavior of solid particle stabilizers (adsorption and desorption) [4]. The wettability, size and interfacial properties of the particles all impact the Pickering emulsion [5]. Figure 1 illustrates the effect of particle hydrophobicity on Pickering emulsions and the structural characteristics of emulsions resistant to coalescence. First, solid particles are irreversibly adsorbed at the interfacial layer, forming a film structure under Brownian motion and capillary pressure, where the physical barrier prevents the merging of droplets together. An increase in the concentration of solid particles results in changes in the structure and viscoelasticity of the film. Bridging occurs when there are not enough particles to cover all droplet surfaces, which does not affect the stability of the Pickering emulsion. Sufficient particles form a 2D or 3D network on the droplet surface to provide space steric hindrance that restricts droplet movement and approach [6].

When the volume fraction of the internal phase (φ) exceeds 74% in Pickering emulsions, high internal phase Pickering emulsions (HIPEs) are formed [7]. φ refers to the internal phase as a fraction of the total volume of the emulsion, and 74% is the maximum fraction for which monodisperse emulsion droplets remain tightly packed without deformation [8]. Due to the special structure and stabilization mechanism, HIPEs are attractive to researchers [4]. For HIPE stabilizers, low usage, high biocompatibility, and a variety of sources are appreciated. The dosage of stabilizers widely used to stabilize HIPEs (φ = 75–90%) is 0.1–2.0 wt%, which is less than the dosage of surfactants [9,10,11]. The critical parameters of stabilizers (wettability, shape, size and surface properties, etc.) are presently under control by various physical or chemical modification methods [5]. Therefore, inorganic particles [12], synthetic materials [13] and natural polymers [14] exhibit potential as sources of HIPE stabilizers. Based on this, food grade particles attract more attention, including proteins, polysaccharides, lipids, flavonoids and other substances [6,15]. For the different base particles involved in the construction of HIPEs, they exhibit various degrees of environmental stability (pH, ions and temperature) and storage stability [16,17,18]. Phase interface films composed of solid particle stabilizers protect the encapsulated unstable substances in HIPEs by physical barriers, including prevention of photodegradation and inhibition of oxidation [19,20]. Moreover, the structural characteristics, digestive features and release mechanisms of HIPEs are closely related to the stabilizer properties and phase composition. The bioavailability of bioactives encapsulated in the HIPEs can be affected by solubility properties, the microstructure of the emulsion, and the stability of the emulsion itself [21]. Firstly, HIPEs possess a relatively high concentration of oil, which means that lipophilic bioactives can be loaded into HIPEs, and a large number of mixed micelles can be generated in the small intestines [22]. Compared with conventional emulsions, HIPEs can attenuate the action of digestive juices and enzymes during the digestive phase of the stomach due to their tight gel structure, which inhibits the aggregation of droplets from forming large droplets and reduces the oxidative degradation of active substances. The stability of emulsions under gastrointestinal conditions affects their digestion rate. HIPEs can break down and release oil droplets, which can then be completely digested by lipase, thus releasing hydrophobic bioactive substances [23]. HIPEs with lignin–chitosan nanoparticles as stabilizers can provide excellent protection for curcumin, while achieving pH-responsive release [24]. Chosen oil phases (hexane or corn oil) can obtain HIPEs for various applications such as oil adsorbents or alternatives for partially hydrogenated oils [25,26]. Protein-based HIPEs means that the protein sources produce particle stabilizers and thus stabilize the HIPEs. Protein-derived Pickering stabilizers are developing rapidly, and the range of protein raw materials is expanding (from plants to animals and even fungi). The regulation of the structural properties of the stabilizer needs to be more precise with the emergence of the new stabilizers [27,28]. Research on the modulation of the structural properties of protein-based HIPEs has gradually matured, and based on their properties, it has been applied to various extents in several fields, in which they can be used as porous material templates [25], in 3D printing ink [29], and as cosmetics [30]. To track these advancements, this review focuses on summarizing the composition and properties of recent protein-based HIPEs. In addition, it provides an overview of the popular protein-based stabilizers and HIPE preparation methods. Finally, it points out the current status and future trends of research on protein-stabilized HIPEs.

## 2. Preparation Route for Protein-Based HIPEs

Figure 2 presents the protein-based HIPEs preparation route. The initial step is to obtain proteins from various materials, either extracted and purified in the lab or by using commercial proteins directly. For instance, commercially available or extracted soybean protein isolates serve as raw materials for HIPE stabilizers [31,32]. At the current stage, the raw material sources of HIPE stabilizers are developed from various plants, animals and fungi, such as peanuts, pork, and bamboo fungus [33,34,35]. Based on the solubility of the target protein, the water-soluble proteins precipitate from the aqueous solution by salting out or from pH-adjustment methods, while organic solvent extraction is used, such as ethanol or acetone, for alcohol-soluble proteins [36]. Different sources and structures of proteins affect the formation and parameters of HIPE stabilizers, but not decisively or exclusively. The transformation of proteins into HIPE stabilizers is the critical step that determines the shape, size and hydrophobicity of nanoparticles [31]. In addition, nanoparticle modifications are sometimes performed to improve the deficiencies in protein stabilizers, such as being susceptible to environmental impacts [37], or they are used to obtain certain performances, including responsiveness and targeting property [38]. The next section describes the detailed mechanisms and methods for the preparation of protein-based HIPE stabilizers.

The next step is to design a reasonable system of HIPEs, including the type and dosage of stabilizer, the type of oil phase and the volume fraction of the internal phase, whether to incorporate other substances, etc. Stabilizer characteristics affect both the type of HIPE (O/W or W/O) and the stability [39]. The dosage of the HIPE stabilizer and volume fraction of the internal phase correlate with the droplet’s distribution pattern and size, which consequently affect their rheological behavior and stability [10,40]. For HIPEs delivering target substances, it is necessary to select a suitable dispersion phase according to the substance’s solubility and usage. Vitamin B_12_ is a water-soluble vitamin dispersed in the aqueous phase of multiple HIPEs (W_1_/O/W_2_), while water-insoluble hesperidin is loaded into canola oil in the oil phase [41,42]. Overall, HIPEs are a complex system of interconnections, and the mechanisms need to be further elucidated in order to design HIPEs that fulfill expectations.

Finally, stabilizers are involved in mixing oil and water phases to form stable HIPEs. In this process, high-energy emulsification is necessary to overcome the system energy barrier and to make the particle distribution reach equilibrium state quickly at the phase interface [5]. Prevalent high-energy emulsification methods include high-speed shear homogenization, high-pressure homogenization, microfluidization, and ultrasonication. Among them, the high-speed shear homogenization provides low equipment requirements, simple operation and high production capacity [43]. For HIPEs with viscoelasticity, high-speed shear homogenization is the more suitable emulsification method, although the performance is poorer than high-pressure homogenization [44]. High-speed shear homogenization is the current mainstream emulsification method used to produce HIPEs [45]. The typical operation is as follows: the protein nanoparticles are dispersed into the water phase and are subsequently emulsified with the oil phase, which constitutes more than 74% of the total emulsion volume fraction in one or multiple steps [26,46].

## 3. Microscopic Characteristics and Stability of HIPEs

As described in the previous subsection, the HIPEs obtained results from a continuous procedure, which means that there are several factors contributing to the microscopic characteristics and stability of HIPEs, including the intrinsic properties of the particle stabilizer (type, wettability, size, shape and geometry) and its concentration, as well as the volume fraction of the dispersed phase and the species used. In addition, other parameters such as the emulsification method and process (shear rate, duration, single or two-step emulsification) can exert an impact in certain situations [47]. There are reports to examine the influence patterns of these parameters as well, but most of them correspond to HIPEs with specific composition, and these discussions only serve as a reference. In general, under the condition that the properties of the solid particle stabilizer and the amount are suitable, a gradual increase in the volume fraction of the oil phase (O/W) can obtain HIPEs with self-supporting properties, the visual appearance of which there is inverted non-flow or that it is not easily deformed, as shown in the Figure 3.

At a certain φ (the exact value is determined by the system), a bridging emulsion is observed, where two oil droplets share the same particle layer, which transforms the emulsion into a gel-like state. Increasing φ further, all oil droplets are interconnected and closely packed, leading to deformation and compression. This restricts the movement of neighboring oil droplets to be close to each other and forms a gel-like network, producing semi-solid colloidal properties of HIPEs [48]. When the particle concentration is fixed, the constant increasing oil fraction relatively reduces the number of solid particles available to stabilize the oil–water interface, leading to the formation of larger droplets until the breakup of the emulsion droplets [49]. Conversely, at constant φ, increasing the concentration of solid particles appropriately allows for faster and closer distribution to the interface, reduced droplet size, and improved structural stability [50,51]. As shown in Figure 4, the oil phase species not only affects the color and texture of the same HIPEs system, but it also varies in its microstructure and rheological properties. Compared with edible oil, the HIPEs stabilized by n-hexane presented higher G′ G″ values, whereas the HIPEs prepared with olive oil presented the lowest G′ G″ values [52]. Therefore, researchers proposed that through changing the oil phase type, viscoelasticity of HIPEs will be controlled and adapted to various application scenarios. In another study, the stability of HIPEs from different oil phases ranked as perilla oil > olive oil > soybean oil > palm oil, which is consistent with the results of the interfacial tension of oils. Due to the variation of properties such as polarity, viscosity, density and fatty acid chain length, different interface structures were formed, while some components of oil were partook in the formation of the interfacial layer, which affected the emulsifying ability of the protein emulsifiers [53,54].

## 4. Fabrication of Protein-Based HIPE Stabilizers

As mentioned above, the stabilizer particle properties dramatically impact HIPEs; therefore, choosing a suitable method to promote proteins that form particles with suitable wettability, size and interfacial properties is required [31,55]. In addition, due to the flexible structure and isoelectric point characteristics of proteins, susceptibility to environmental influences (pH, and ionic strength) leads to changes in the structure and stability of HIPEs [28,56]. Some studies modify the protein stabilizer to attenuate this impact, by using other complex substances (polysaccharides, polyphenols, etc.) during or after protein particle formation [57,58].

There are two routes to prepare the stabilizer, top-down and bottom-up, based on the change in size during the particle preparation process. Top-down, which means to first obtain large particles and then reduce the particle size by shearing or ultrasound until it meets the requirements, is commonly used in the preparation of stabilizers based on the protein gel method [31,42]. Conversely, bottom-up refers to the aggregation or precipitation of proteins under controllable solvent conditions, resulting in the particle stabilizer [16,59]. In more detail, as shown in Figure 5, proteins consist of polypeptide chains with multiple interaction forces and functional groups, which contribute to forming diverse protein particles. According to the dominant reactions during the formation of protein-based HIPE stabilizers, Figure 6 shows the various methods and mechanisms for protein-based stabilizer preparation.

### 4.1. Non-Covalent Interactions 

Non-covalent binding uses electrostatic interactions, hydrogen bonding, and hydrophobic interactions, which make the protein combine with itself or different substances, consistent with the definition of self-assembly [60]. The anti-solvent method is commonly used to prepare HIPE stabilizers from alcohol-soluble proteins based on self-assembly. The alteration of solution polarity induces the protein solubility change, which leads to a molecular conformation transformation, resulting in the particle formation. The general process is summarized as follows (Figure 6a): The proteins (gliadin, zein, and hordein) are dissolved in a suitable solvent, typically an ethanol–water system; this solution is then added dropwise to the aqueous solution. The solution polarity alters, protein solubility decreases, precipitation and self-assembly occur, and finally, a solution containing a particle stabilizer by removal of ethanol is obtained [59,61,62]. In addition to protein self-assembly, the anti-solvent method can prepare protein-based particles. The gliadin solution is added dropwise to the chitosan dispersion and is pH adjusted to obtain gliadin–chitosan complex particles, similarly to zein–pectin composite nanoparticles, where electrostatic adsorption and hydrogen bonding interactions play essential roles [25,59].

The pH adjustment method is also a HIPE stabilizer preparation method based on non-covalent interactions (Figure 6b). Regulation of the solution to extreme pH (acid or base) away from the protein isoelectric point results in an increase in the charge carried by the protein side chains and an increase in electrostatic interaction forces. Meanwhile, the protein molecular structure undergoes unfolding, and the internal groups are exposed, facilitating the binding with substances [63,64,65], which is suitable for water-soluble proteins to prepare the stabilizer. At pH 3.0, ovalbumin aggregates form spherical particles with sizes of around 90 nm; a 0.07 wt% particle can stabilize gel-like HIPEs [55]. When oppositely charged polyelectrolytes are present in a same solution, electrostatic interaction might occur and lead to the formation of insoluble particles [18]. Tuning the pH of the solution allows for the formation of various complex particles, including protein–polysaccharide, protein–protein, and protein–polyphenol complex stabilizers. Continuous stirring of the pea protein isolate–high methoxyl pectin–epigallocatechin gallate (EGCG) complexes formed complex particles at pH 3.5, which exhibited partial wettability (81.6 ± 0.4°) to stabilize HIPEs [66]. The properties of the cationic lactoferrin and anionic gum Arabic complex particles change by modulating the pH of the solution, including particle size, turbidity, zeta-potential, etc. [20]. 

### 4.2. Covalent Interactions

A protein is composed of amino acids and contains a variety of molecular functional groups (amidogen, carboxyl, sulfhydryl, etc.), which is very suitable for constructing covalent bonds. Recent reports on the preparation of HIPEs frequently involve the gelation method [22,33,38]. According to the reaction conditions, gelation includes heat-induced and cold-induced gelation (Figure 6c), the latter of which can be subdivided into enzyme-induced, salt-induced, and acid-induced [67]. Among them, enzyme-induced and heat-induced gelation are treatment methods based on covalent action (heat treatment is related to sulfhydryl groups). For whey protein, the stability of HIPEs prepared after heat-induced gelation is significantly improved compared to that of ungelatinized whey protein and small-molecule surfactants [68]. Under mild conditions (45 °C), peanut-protein gels are formed between molecules or within molecules by acyl-transfer reactions initiated by enzymes, which stabilize HIPEs with up to 88% oil phase volume fraction [33]. 

The amino group of the protein and the reducing-sugar carboxyl group will undergo a Maillard reaction under heating conditions, which is common in food processing [69]. The reducibility of the products in the early stage of the reaction is enhanced, and the toxicity, flavors, and color do not change. Under artificially controlled conditions, binding proteins and polysaccharides through the Maillard reaction is a green and simple-operation method (Figure 6d) [70]. The combination of hydrophobic protein and hydrophilic polysaccharide is prevalent to attenuate the effects of pH and ionic strength on the HIPE structure. Compared with bovine serum albumin (BSA) particles, glycated BSA is more inclined to form HIPEs, which display improved performance during long-term storage, heating and freeze–thawing [71]. Glycated soy glycinin increased the gel network strength while reducing the concentration of the stabilizer required, owing to the high structural stability of particles at the interface [9]. Similarly, the thermal and oxidative stability of HIPEs stabilized by glycated pea protein particles was enhanced [39].

Glutaraldehyde is the primary cross-linking agent and cross-links with amino groups in and between protein molecules [72]. Type B gelatin undergoes intramolecular cross-linking to form gelatin nanoparticles through glutaraldehyde cross-linking. This kind of particle can not only control the release behaviors of β-carotene from HIPEs by adjusting the concentration but can also construct a degradable 3D scaffold for cell growth [73]. In addition, free radical-induced reaction-generated whey proteins–EGCG conjugate nanoparticles with better wettability and smaller particle size [74].

## 5. Critical Parameters of Stabilizers

The formation and stabilization of HIPEs depend on the interfacial properties of solid particle stabilizers, such as the adsorption, distribution, and assembly behavior at the phase interface, the ability to reduce the interfacial tension, the network formation at the oil–water interface, the thickness of the adsorption layer, and the interaction between the adsorbed particles [4]. The particle size of the stabilizer particles is a key factor in determining whether a Pickering emulsion system can be constructed. Regardless of which material is chosen to prepare the particles, the purpose is for the particles to adsorb at the oil–water interface to achieve the effect of stabilizing the emulsion; thus, the size of the particles should be much smaller than the droplet size of the Pickering emulsion when the particles are processed. Proper wettability and particle size make the particles irreversibly adsorbed at the oil-water interface, forming a spatial potential barrier that prevents the aggregation of the dispersed phase. The wettability, shape, and size of the particle stabilizer are generally discussed in studies. The particle stabilizers must be partially wetted by both the aqueous and oil phases while keeping intermediate wettability to achieve sufficient interfacial absorption. The three-phase contact angle reflects wettability, and in general, particles with contact angles in the range of 15° < θ < 90° lend to stabilizing O/W emulsions. In the range of 90° < θ < 165°, a W/O emulsion is the preferred form. The θ of protein–polysaccharide conjugates extracted from shiitake mushroom were 115.0, 118.8, 92.1, and 75.5 at pH 3, 5, 7, and 9, respectively, where particles with contact angles of 115.0 and 118.8 failed to construct stable O/W HIPEs (φ = 78%) [27]. However, few stabilizers in Table 1 seem to deviate from this pattern, as they show θ > 90°, yet they formed stable O/W HIPEs [29,48]. This indicates that the three-phase contact angle is not the unique determinant for the type of emulsion. In addition, from an energy perspective, the energy required for spherical particle desorption from interfaces is calculated as
ΔG = πR^2^γ(1 ± cosθ)^2^
where R is the spherical particle radius, γ represents the oil–water interfacial tension, and θ is the three-phases contact angle. When θ approaches 90°, the particles are both hydrophilic and lipophilic, the adsorption energy of the interface to the particles is at the maximum, and the formed droplets are more stable, which is an ideal Pickering stabilizer [75,76]. Pickering stabilizers with various shapes and anisotropy correspond to different equations [2].

The morphology of the particles determines their adsorption behavior at the interface and thereby determines the stabilization capacity of the emulsion [96]. With the same volume of substances, spherical particles have the smallest surface area and the lowest total Gibbs free energy on the surface. At this time, the particles are easily adsorbed to the interface and are packed tightly in a hexagonal shape, which is conducive to the reduction of the interfacial tension [97]. As shown in Figure 7b, protein-based Pickering stabilizer shapes also include spherical particles; for instance, spherical hordein particles with a diameter of about 200 nm adsorbed at the O/W interface form a relatively thick layer of hordein nanoparticles around the oil droplets, as well as aggregation in the continuous phase [42]. Similarly, Figure 7b shows that regular elliptical β-lactoglobulin nanoparticles are formed by chemical cross-linking, of which the stabilized HIPEs exhibit thermal and freeze–thaw stabilization benefits [98]. Figure 7d reveals that whey isolated protein nanofibrils with long and semi-flexible structures formed by self-assembly can stabilize HIPEs (φ = 90%) [85]. In addition to the common regular shapes, irregular and special shapes such as hollow ring-shaped particles exist as protein-based Pickering stabilizers [50,99,100]. Inorganic particle stabilizers have diverse shapes as well, but in contrast, protein-based particles are soft particles [71,101]. This means that its structure and shape may undergo variations within a limited extent when the environment changes or during the adsorption process. Soft particles exhibit faster interfacial adsorption rates, higher interfacial coverage and the superior ability to reduce interfacial tension compared to rigid particles; Figure 7a reveals this situation from a microscopic perspective [102]. 

Droplet size is a microscopic characteristic, and microscopic properties play an important role in the study of Pickering emulsions. For Pickering emulsions, droplet size is generally influenced by particle type, particle concentration, oil and water volume fraction, etc. Single globular proteins change the droplet size of the stable Pickering emulsion after the addition of tannic acid [104]. Pickering emulsions stabilized with alcohol-soluble particles of different concentrations of maize protein showed the same variability in droplet size after preparation [105]. For Pickering emulsions, the higher the volume fraction of the oil phase, the larger the droplets will be [106]. 

Based on the adsorption energy equation, larger particles adsorbed on oil droplets require more energy to fall off from the interface, while larger particles are more tightly adsorbed at the interface, causing a stronger interfacial structure. For protein particles with diameters in the range of 120–350 nm obtained by the anti-solvent method under different conditions, the size of the emulsion is smaller when formed by particle stabilizers with larger diameters. Meanwhile, small particle-stabilized Pickering emulsions become smaller droplets during storage due to the relatively low desorption energy, resulting in particle separation from the interface. This process also causes changes in apparent viscosity [107]. However, the adsorption kinetics of larger particles is slower, and the adsorption efficiency at the interface is lower. As shown in Figure 8, smaller particles diffuse more easily on the droplet surface and rapidly and spontaneously form an orderly arrangement at the interface [5]. Pickering emulsions stabilized by silica particles with an average particle size of 68 nm reached a stable state more quickly and provided superior stability compared to large particles [108]. For myofibrillar gel particles (100–250 nm), their size did not affect the Pickering emulsion droplet size, which the authors consider might result from the fact that the turbulence generated by the homogenizer during the emulsion preparation is much faster than the spontaneous adsorption of the particles, thus shielding the effect of particle size [109]. The correlation between the particle stabilizer and droplet size and stability is complex, but particle size must be at least one order smaller in magnitude than droplet size, and the data in Table 1 with Table 2 are consistent with this trend [75].

## 6. Exploration of Protein-Based Stabilizers

### 6.1. Plant Proteins 

Zein, with an average molecular weight of about 44 kDa, is the primary storage protein of corn. α-zein and β-zein have been directly extracted with 90% (*v/v*) 2-propanol and are the main components of zein [111]. Nanoparticles prepared from zein alone serve as Pickering stabilizers by anti-solvent methods [112]. However, the complex zein particles are preferred in HIPE stabilizers, where polysaccharides, phenols, and lipids all bind to it. Non-covalent conjugates composed of zein and tannic acid resulted in well stabilized HIPEs. Adjusting the pH of the solution during the binding process helps to achieve control of the HIPEs’ rheology [46]. Loading curcumin to zein and pectin self-assembled particles stabilized HIPEs, reduced the UV-induced photodegradation of curcumin, and exhibited excellent storage stability [77]. HIPEs stabilized by zein, propylene glycol alginate, and rhamnolipid complexes exhibited good physical stability, which can withstand drastic temperature changes [78]. Different from the direct method that combines with zein to prepare particle stabilizers, complexation with zein particles during the emulsification of HIPEs has been reported. After emulsifying together the starch nanocrystals and the zein nanoparticle system, a bilayer interfacial film was formed, which synergistically improved the stability and resulted in smaller and more uniform droplets of the HIPEs [113]. Edible HIPEs with double-emulsion morphology (W_1_/O/W_2_) were prepared from zein particles and soy lecithin, showing potential applications in nutrient delivery [79].

Wheat proteins include albumin, globulin, gliadin, and glutenin. Among them, gliadin and glutenin account for 80% of the wheat seed proteins [114]. There are four categories of gliadin based on differences in electrophoretic properties: α-gliadin and β-gliadin, with average molecular weights of about 31 kDa, and γ-gliadin and ω-gliadin, at 35 and 40–75 kDa, respectively [115]. Wheat protein preparation for particle stabilizers is commonly used during the anti-solvent method. Wheat protein and polysaccharide complex particle stabilizers occupy a major portion, including chitosan, gum Arabic, etc. The complex stabilizer of gliadin and chitosan improved the performance of HIPEs. Specifically, after four weeks of storage at room temperature, there was no significant change in particle size, and the lipid peroxides in the oil phase were inhibited. After loading curcumin, the HIPEs restricted the ingestion of oil and enhanced the bioavailability of curcumin from 2.13% (bulk algal oil) to 53.61% [26,80]. The gliadin particles prepared by the anti-solvent method bind with gum Arabic to form a composite stabilizer using the pH adjustment method. The composite particles constitute a compact 3D network in the HIPEs, making it relatively stable to changes in pH, ionic strength and temperature [37]. According to a similar process, the binding and modification of wheat protein with polyphenols was also implemented. Rice bran-modified wheat gluten nanoparticles were adsorbed at the O/W interface, forming an interconnected network structure and depressing the oxidation of soybean oil [48]. 

Glycinin and β-conglycinin from soybeans were the main components of 11S and 7S, respectively, with average molecular weights of 312 and 180–210 kDa, the total of which accounted for about 70% of the soybean’s protein [116,117]. Soybean isolated proteins are a classical source of stabilizers for HIPEs, and studies on their classification, preparation and modification are more diverse compared to other proteins. The synthesis of soy nanoparticles includes heat induction, Maillard reaction with dietary fiber, anti-solvent binding with cellulose, or covalent conjugation with anthocyanins [57,82,118,119]. HIPEs stabilized by heat-induced unmodified soy protein microgels have a slower free fatty acid release profile in intestinal digestion in vitro [31]. Complexes of Heated soy protein isolate and chitosan at pH 3.0 as the stabilizer, showed a high capacity to stabilize HIPEs with a high freeze–thaw stability [120]. With more subdivision, soy β-conglycinin could perform as an outstanding Pickering stabilizer, forming HIPEs that are resistant to heat (100 °C for 15 min) [40]. Heat treatment induces the formation of soy glycinin–soy soluble polysaccharide gel particles, and then, they are emulsified with soybean oil to form HIPEs [9]. All the acid-treated soybean lipophilic proteins, β-conglycinin, and globulin have succeeded in forming stabilized HIPEs at pH 2.0 [10].

Besides the three major types of grains mentioned above, protein-based HIPE stabilizers obtained from peanuts, peas, etc., are also receiving attention. Peanut proteins commonly employed for the preparation of Pickering stabilizers by gelation methods in reports include enzymatic cross-linking as well as thermal and Na^+^-induced aggregation [33,121]. Peanut protein gel particles undergo different aggregation forms at different pH levels, and their stable HIPEs exhibit multiple application potentials in food and material fields [33]. Rice protein, which is hypoallergenic, has a nutritional value equivalent to egg and milk proteins. By co-assembling rice protein and carboxymethyl cellulose as HIPE stabilizers through the pH adjustment method, the HIPEs show favorable printing resolution, hardness, and adhesion as a 3D printing ink [29]. Similarly, the pea protein isolate–high methoxyl pectin–EGCG complexes can stabilize HIPEs [66]. Recently, more and more plant proteins have been successfully utilized to construct HIPE stabilizers, such as pecan protein, hordes, and quinoa proteins [28,62,122]. Researchers have noticed not only the high nutritional value of seeds and leaves, but also the proteins in processing waste. Proteins derived from perilla oilseed residues undergo pH adjustment and homogenization to produce particles of 224 nm. HIPEs (φ = 75%) constructed with 1% stabilizer are extremely stable for long-term storage, heating and ionic strength, but are susceptible to freeze–thaw treatment [123].

### 6.2. Animal Proteins 

Whey protein, a globular protein primarily composed of α-lactalbumin and β-lactoglobulin (two small globular proteins accounting for approximately 70–80% of total whey protein) are commonly available in food [124]. There are several studies on preparing HIPEs using whey protein as stabilizers, and gelation is the primary way to prepare nanoparticles. Although the methods of inducing gelation are different (calcium ions, heat treatment or cold-set gelation), all of the resulting HIPEs exhibit excellent stability and the ability to deliver hydrophobic substances (β-carotene), both reducing their degradation and increasing their bioavailability [22,38,90]. In addition, complex particles of whey proteins are also available as stabilizers for HIPEs, including whey protein–low methoxyl pectin, and whey protein–EGCG complexes [74]. Development of whey protein stabilizers and property modulation of HIPEs based on different preparation methods have made considerable progress. During the gelation process, Ca^2+^ concentration plays a dominant role in nanoparticle formation and characteristics (Z-average diameter and zeta-potential values); the high Ca^2+^ concentration leads to a multimodal particle size distribution of particles [38]. Control of whey protein concentration leads to gel particles with different rigidities, which differ in their adsorption behavior at the interface. Soft microgel particles (5%) show excellent deformability, faster interfacial adsorption rates, and higher interfacial coverage compared to harder microgels (10% and 20%). A soft particle stabilizer applied in the form of a HIPE as a healthier margarine substitute showing a delayed lipid digestion rate [102].

BSA is composed of 583 amino-acid residues with a molecular weight of 66.3 kDa. It is the main protein in bovine serum and is frequently used in biochemical experiments [125,126]. After simple mixing, BSA is bound to sucrose by electrostatic action at a specific pH. The presence of sucrose transforms BSA into soft nanoparticles with high structural integrity, ultimately promoting the formation of HIPEs [87]. A more complex approach to obtaining protein–polysaccharide complexes is the production of covalent cross-links between BSA and galactose through glycosylation. These protein–polysaccharide particles can significantly improve the emulsification capacity and stability of the HIPEs compared with natural BSA [71,127]. BSA and carboxymethyl cellulose bonded by the pH adjustment method, which controls their ratio, can regulate the stiffness and structure of HIPEs [128].

Gelatin is a product obtained by the hydrolysis of animal collagen. According to different treatment methods, gelatin includes A-type (acid treatment) and B-type (alkali treatment). Gelatin has a molecular weight of 50–100 kDa and is widely used in the food and pharmaceutical industries. Chemical modification had mainly obtained gelatin nanoparticles in the reports of HIPEs [54]. Acetone induces gelatin to form high-molecular-weight gelatin, and glutaraldehyde, a cross-linking agent, is added to produce gel particles [129]. Altering the concentration of gelatin nanoparticles can control the structure and rheological properties of HIPEs. With high particle concentrations, the HIPEs become more solid-like, the particle size is smaller, and it can delay the release of functional factors in the oil phase [95]. Removal of the dispersed phase of the HIPEs can produce protein scaffolds with high porosity, smooth pore walls, and a textured structure. The protein scaffolds are degradable and have good biocompatibility [73]. Although the obtained protein scaffolds are degradable, the removal of chemical cross-linkers is an issue worth noting. The adoption of green cross-linking agents such as genipin or by altering the method of synthesizing gelatin Pickering stabilizers will be readily accepted. Gelatin–catechin self-assembled nanocomplexes might serve as excellent food-grade Pickering emulsifiers for enhancing the interfacial and antioxidant properties of HIPEs (φ = 80%).

Ovalbumin, a water-soluble protein, is the main component of egg white protein, accounting for 54–65%. It does not have inherent high surface hydrophobicity, unlike BSA. Due to hydrogen bonding and hydrophobic interactions, the wettability of ovalbumin–tannic acid composite particles improved significantly, resulting in stable HIPEs with a narrow size distribution, compact droplet and gel network structure [89]. The thermostabilized form, S-ovalbumin, has higher surface hydrophobicity, zeta-potential, and a thicker water layer than the ovalbumin. The HIPEs obtained based on S-ovalbumin have a firm network structure, showing the inhibition of lipid oxidation [88]. Lysozymes derived from egg, a multifunctional natural single-chain protein with 129 amino acids, mixed with dihydromyricetin stabilizes HIPEs with a volume fraction of 90% [130]. In addition, there are more protein types present in eggs that have the potential to act as HIPE stabilizers. The low-density lipoprotein in egg yolk was used as a stabilizer for HIPEs with a particle size of 49 nm and a three-phase contact angle close to 90°. Stabilized HIPEs (φ = 80%) show thermal and freeze–thaw stability, while providing photoprotection of the delivered curcumin and improving its bioavailability [131]. While egg yolk high-density lipoprotein also stabilizes HIPEs, its structure and properties are sensitive to pH [132]. A recent study successfully prepared HIPE stabilizers from the low-value product, salted duck egg white. First, the egg white was converted into hydrogel via the hydrothermal method, and then, it desalted and broken by ultrasound to obtain the low-cost but efficient food-grade Pickering emulsion stabilizer [133].

Daily recipes for adults should include the four main categories of dairy, meat, vegetables and fruits, and grains. Coincidentally, current studies of stabilizers of HIPEs all involve these categories. Lactoferrin is a globular glycoprotein containing around 700 amino acids (80 kDa) isolated from bovine milk. The lactoferrin–gum Arabic complex formed by electrostatic interaction as the HIPE emulsifier improves the photostability of curcumin [20]. Meat protein (pork) particles enable the form of HIPEs in a wide pH range (3–11) and exhibit good stability, including heating, freeze–thawing, and storage [34]. Further, the microstructure of myofiber obtained from cod meat is affected by pH. For example, HIPEs prepared at pH ≥ 7 have better apparent mechanical properties for 3D printing, oil adsorption, and cell scaffolds [93]. Stabilizers obtained by enzyme-induced sea bass protein gels had pH-dependent particle sizes with better stability and wettability at pH 7. The HIPEs stabilized by microgel particles could be used as a delivery vehicle for astaxanthin for edible functional foods [94].

### 6.3. Fungal Proteins

Compared to the diverse protein stabilizers from plant and animal sources, fungus is still under-exploited as another broad protein source with great potential in HIPE stabilizers. Edible fungus contains nutrients such as polysaccharides, proteins, terpenoids, minerals and vitamins, among which proteins and polysaccharides used to develop Pickering emulsion stabilizers can realize the full utilization of fungal resources or obtain high-value products [134,135]. Previous studies extracted proteins from bamboo fungus fruit entities and enzymes induced to produce gel particles for HIPE (φ = 80%) stabilization, which destabilizes at pH 5 and 7 only; otherwise, it exhibits excellent storage stability after 5 months of refrigeration [35]. Protein–polysaccharide conjugates extracted from mushroom roots have a bead-string nanostructure with a bead diameter of 300 nm and a string thickness of 50 nm. Their contact angle varies with pH value, from 75.5° to 115°. HIPEs prepared at pH 7 have good long-term storage stability (30 days) and heat resistance (90 °C), which can be used for 3D printing to produce various objects with strong self-supporting and high structural resolution properties [27]. Table 1 summarizes the critical parameters related to the particles produced from the above protein sources.

## 7. Other Ingredients in Protein-Based HIPEs

In a well-designed emulsion system, besides the three primary agents including emulsifiers, water, fat or oil, texture modifiers and several other food additives such as antibacterial agents, antioxidants, colors, and flavors are also involved [136,137]. However, protein-based HIPEs are currently in the basic study stages, without a complex system composition, mainly comprising stabilizers, water, oil and delivered substances. According to the preparation route of the HIPEs shown in Figure 2, it directly disperses the protein particle stabilizers in water and then co-emulsifies it with the oil phase and the delivered substance. Stabilizer details are discussed above in the article; therefore, the following describe the oils and delivered substances in protein-based HIPEs, the details of which are summarized in Table 2.

### 7.1. Oil Phase

The type of oil phase significantly determines the application of HIPEs, and the physical properties of the oil phase (polarity, viscosity, etc.) affect the structure and stability of the HIPEs [141]. In emulsions where the oil phase crystallizes after the aqueous phase, the instability of the emulsion will be influenced by the crystallization pathway of the oil rather than by the type of oil. The smaller the difference between the storage temperature and the melting point of the oil phase (as estimated from the fatty acid composition), the more likely it is that the instability of the emulsion will occur prior to the crystal growth of the oil phase. The crystals of the oil phase are more likely to destabilize the emulsion prior to crystal growth. Based on the crystallization pathway of the oil through the melt, it is shown that the energy at the interface between oil and water may influence the instability of the emulsion [142]. The difference in melting temperature of palm oil in black and white polyemulsions stabilized with beeswax was attributed to the variation in its fatty acid composition. The size and morphology of oil droplets in film-forming emulsions varied with the palm oil content [143]. 

#### 7.1.1. Hydrocarbons

Currently, oil phases in protein-based HIPEs comprise two main categories: hydrocarbons and edible oils. As a common extraction solvent, N-Hexane appeared previously as the oil phase in reports on protein-based HIPEs. Besides meeting the basic requirements for forming stable HIPEs (insoluble with water), volatilization easily removed n-hexane from the HIPEs because of its low boiling point. Thus, the HIPEs were freeze-dried after complete removal of the oil phase to obtain intact protein porous material [73]. In addition, dodecane plays a similar role in the formation of stable HIPEs. Specifically, squalane, a precursor substance for steroid synthesis, takes part in metabolism and is a kind of high-purity oil compared with vegetable oils. This is encapsulated in an HIPE as an oil phase with a double emulsion structure (W_1_/O/W_2_), showing potential applications in nutrient delivery [79].

#### 7.1.2. Edible Oil

Using protein particles instead of a surfactant as a stabilizer allows for the preparations of green, non-toxic, and biocompatible HIPEs. Edible oils are ideal oil phase components in the food field, consistent with this concept. Four edible vegetable oils (soybean oil, peanut oil, olive oil, and sunflower oil) have been prepared as HIPEs with zein–starch stabilizers to explore the feasibility of replacing traditional mayonnaise and salad dressings [113]. Similarly, the conversion of liquid corn oil into solid viscoelastic emulsion gels based on HIPEs is a suitable substitute for solid fats to achieve no intake of trans fat and saturated fat [61]. The effect of oil phase properties is noticeable in the structure’s investigation and properties of emulsions. Increasing the amount of canola oil in the dispersed phase leads to an increase in the apparent viscosity and elastic properties of the Pickering system [62]. Ovotransferrin–gum Arabic-particle stabilized HIPEs improve the lipolysis of the medium-chain triglyceride (MCT) oil compared with bulk MCT oil [144]. Using HIPEs to improve the application defects of functional oils is also receiving attention. HIPEs comprising tea tree oil (6.0 wt%) and jojoba oil as the oil phase significantly reduced the growth of *Staphylococcus aureus* and *Escherichia coli* with inhibition rates of 80% and 95%, respectively [145].

### 7.2. Encapsulated Substances

Many water-insoluble substances, such as curcumin, have significant physiological activity and various health benefits. Still, their bioavailability is very low due to their unstable physical properties or poor absorption in the gastrointestinal tract, limiting their application and constructing HIPEs as a delivery system for curcumin and effectively improving the bioavailability of curcumin [146]. It is an excellent way to disperse these substances into the oil phase in HIPEs, which would protect the bioactive substance from UV light, and improve the thermal stability in vitro and the stability in gastric digestion [10,26]. HIPEs interfacial film composed of pecan protein–xanthan gum complex particles contributes to the high retention of quercetin at high temperature, ferric ions and hydrogen peroxide. The interfacial film produces a barrier action that prevents the entry of effective oxygen, while the gel-like network structure delays the direct contact between quercetin and ferric ions and prevents their degradation [122]. Attributed to the structure of HIPEs, the higher concentration of stabilizers has a higher retention of astaxanthin in HIPEs stabilized by sea bass protein microgel particles. Meanwhile, the bioavailability of astaxanthin during in vitro digestion was improved to 51.17% [94]. The gel network in HIPEs inhibits the coalescence and flocculation of oil droplets. The droplets have a large surface area, which increase the access of lipase and bile salts to the surface of the oil droplets and increase the transfer of encapsulated substances from the oil phase into micelles, ultimately improving bioavailability [38]. Herein, many active substances are added to protein-based HIPEs to design more stable and easily absorbed dosage forms, such as β-carotene, indomethacin, flavonoid, luteolin, hesperidin, etc. [10,41,92]. Unlike inert inorganic particles, proteins stabilizers are prone to undergo chemical reactions with substances in protein-based HIPEs system, which require further consideration. Co-oxidation of proteins and oils at the interface is an ongoing and complicated study, involving different protein and lipid combinations to promote/inhibit oxidation [147]. During the digestion of the protein-based HIPEs, the interfacial films composed of protein stabilizers are vulnerable to digestive enzymes, and the above structure of HIPEs may be destroyed, depending on the properties of the different stabilizers, which will affect the bioavailability of the substance and oil digestion [38,131].

## 8. Application in the Food Field

### 8.1. Fat Substitutes

The high levels of saturated fats in animal fats are in relatively limited supply; thus, there are more studies trying to find a breakthrough in the field of vegetable oils, which can increase the plasticity of fats through the process of hydrogenation, but this can lead to the production of harmful trans fats. Both saturated and trans fats are considered unhealthy because their intake increases the risk of cardiovascular disease [26]. As a result, the food industry has placed great emphasis on identifying healthier alternatives that provide the same desirable textural properties typically associated with saturated and trans fats, and Pickering emulsions have been shown to be used in the manufacturing of a variety of foods as fat substitutes. These semi-solid fats can form 3D networks of aggregated fat crystals, which contribute to the unique textural properties of foods such as margarine, sauces and baked foods [61].

Fats are usually digested slowly and are easily oxidized; moreover, in food applications, lipid oxidation can adversely affect the quality of food products. The microstructure of Pickering emulsions at the oil–water interface can effectively reduce lipid oxidation. Pickering emulsions are more advantageous in the oil digestion process and in reducing lipid oxidation. The selection of suitable particles as picolin stabilizers can regulate the rate and extent of lipid digestion. Previous studies found that oil gel-based Pickering emulsions stabilized by ovotransferrinogen fibrils had higher digestion rates and greater lipolysis than oil gels [131]. 

Ethyl cellulose-stabilized Pickering emulsions can replace cream in the preparation of frozen yogurt and ice cream. Sensory evaluation and physicochemical characterization showed that the food quality of Pickering emulsion-based products was more satisfactory. Pickering emulsions can also be used as a substitute for butter in the preparation of cakes. Pickering emulsions produce cakes with reduced calorie intake and longer shelf life without changing the color and texture [112]. The high levels of saturated fats in animal fats are in relatively limited supply; thus, there are more studies trying to find a breakthrough in the field of vegetable oils, which can increase the plasticity of fats through the process of hydrogenation, but this can lead to the production of harmful trans fats. Both saturated and trans fats are considered unhealthy because their intake increases the risk of cardiovascular disease. Pickering emulsions offer a good alternative to fat substitutes. 

### 8.2. Delivery of Nutrients

Nutrients such as curcumin, hesperidin and β-carotene, which play a beneficial role in human health, are often made into health foods in the food sector, but most bioactive compounds are susceptible to oxidation and decomposition; thus, hydrophobic bioactives must usually be encapsulated to increase their dispersibility, stability and bioavailability [84]. In contrast, Pickering emulsions are considered as a potential delivery system for fat-soluble bioactive ingredients, with higher bioavailability of nutrients when loaded into Pickering emulsions, and in Pickering emulsion systems, solid particles adsorbed at the oil–water interface can provide a physical barrier, thus weakening the degradation of nutrients. Starch-stabilized HIPEs modified by octenyl succinic anhydride (OSA) have been shown to improve the photostability of β-carotene [22]. Similarly, HIPEs stabilized by a complex of pea protein isolate with high methoxylated pectin have been shown to improve β-carotene bioavailability [84]. Similarly, HIPEs can enhance the chemical stability of bioactive oils. Some polyunsaturated fatty acids (PUFAs) have many health benefits, but they are susceptible to oxidation at high temperatures, oxygen and light, leading to undesirable sour odors [148]. Algae oil is rich in omega-3 PUFAs and is a viable alternative to fish oil in vegetarian and vegan diets. However, it is often difficult to incorporate into functional foods due to its propensity for rapid oxidation [149]. HIPEs allow a thick protective layer to form around the oil droplets to achieve oxidation inhibition, spatially preventing pro-oxidants from reaching the PUFAs, as well as the natural antioxidant properties of some of the components used to assemble the emulsifiers. For example, HIPEs prepared from OSA starch–chitosan complexes as stabilizers have been shown to inhibit the oxidation of algal oil [150]. 

### 8.3. Detergent

In the food sector, the production and processing of food companies inevitably produce a large amount of oil and grease, and not only that, but in everyday life, catering brings a large amount of residual oil stains. Many countries around the world have established food safety standards for detergents, indicating that detergents are closely related to food safety and food science [151]. Conventional detergents consist mainly of surfactants. Daily use of such detergents can have a negative impact on the environment, and the safety of traditional chemical fungicides and preservatives in vegetable preservation is increasingly questioned by consumers, as the widespread use of these chemicals has led to resistance to *E. coli* O157:H7 [152]. Compared with traditional synthetic surfactants, the emulsifiers of Pickering emulsions are mostly biological macromolecules, such as proteins or polysaccharides, which have the advantages of being green, effective and safe [153]. For example, composite colloidal nanoparticles were prepared from wheat gliadin (Gl) and soybean polysaccharides (SP), and their stabilized Pickering emulsion could be used as a detergent, and the colloidal nanoparticles were stabilized by adsorption at the oil–water interface, forming a thicker interfacial film. Pickering emulsions inhibited the growth of *E. coli* O157:H7 on the surface of fresh cabbage while maintaining chlorophyll quality, chlorophyll content and sensory perception of the fresh cabbage [154]. Detergents made from solid particles are not only environmentally friendly, but they also provide better stain removal.

### 8.4. Assistance with Other Delivery Systems

Microspheres with small size and volume, large specific surface area, good diffusion and dispersion ability, uniform or variable size, special surface chemistry, and novel morphology provide good facilities for different fields. Thus far, many preparation methods have been proposed, such as emulsion methods (single and double emulsion methods), phase separation, spray drying, supercritical fluid method, etc. In addition to the above mentioned methods, Pickering emulsification technology is an attractive option due to its stability, feasibility, and homogeneous product size [155]. In recent years, micro-scaffolds have also been a hot topic of research due to the necessity of bone tissue engineering. Pickering emulsion technology, including in situ gelation of alginate, was applied to fabricate porous scaffolds for dual drug release and tissue engineering. According to their drug release results, hydrophobic ibuprofen (IBU) has a good release profile, while bovine serum albumin (BSA) has a faster cumulative release rate due to the two-stage release of internally contained IBU and the hydrophobic difference between the two drugs [156]. 

Microcapsules can basically be referred to as microspheres with cavities. Microcapsules can likewise be prepared by picolin emulsions, and it has been reported that pre-MF prepolymers (pre-MF) can polymerize in situ to form composite shells after absorption of hap-stabilized picolin droplets, and microcapsules are formed after solvent evaporation [157].

Janus colloidal particles usually have two different chemical properties at the same time, such as hydrophilic on one side and hydrophobic on the other, or they can have a new non-spherical shape. Pickering emulsions are considered to be one of the most efficient intermediates for the preparation of Janus colloidal particles, and the basic process is usually a chemical modification or etching at the oil–water interface of the Pickering emulsion [158]. Amphiphilic titanium dioxide nanoparticles, containing phosphonate anchor groups, were prepared at the oil–water interface by adding a hydrophobic coupling agent to the oil phase and a hydrophilic coupling agent to the water phase. Wettability tests were performed on titanium dioxide nanoparticles. Due to isotropic nanoparticles, better wettability properties were obtained in both water and oil phases [159]. Pickering emulsions can be a good choice for the surface modification of nanoparticles.

## 9. Future Perspectives

In short, a wide source of protein Pickering stabilizers and a variety of preparation and modification methods are now available [4]. There are systematic theories and explanations for the factors and mechanisms affecting the stability of protein-based HIPEs, and the newly emerged kinds of HIPEs still require specific exploration [75]. Until now, the potential of protein-based HIPEs with different compositions in food has been proven [160]. However, these studies were in the experimental stage and were insufficient, and they still need exploration. 

### 9.1. Development of Protein-Based Stabilizers 

As shown in Table 1 above, the range of proteins used as resources for stabilizers for HIPEs is extensive and completely covers plants, animals, and fungi. The variety of proteins that serve as a resource for HIPE stabilizers is relatively inadequate. There are just a few types of proteins that are commonly used, including zein, soy protein, gliadin and whey protein. Their costs need to be considered, especially for gliadin and whey proteins. It is meaningful work to seek low-priced, high-content protein resources from large quantities of production byproducts or scrap material. Whether the nutritional value of protein is meaningful for HIPEs is an undecided issue, as the dosage of stabilizers is too low. Differently, protein-based stabilizers with functional activity are worth consideration, such as anti-oxidation or improving the oxidative stability of delivered substances through multiple pathways, for instance, the selection of protein or stabilizer modifications. Selecting the suitable preparation method for particle stabilizers according to the hydrophilic properties and functional groups of proteins is promising. Modification of established protein particle stabilizers to overcome defects or add functionalities contributes to the full exploration of stabilizers for HIPEs.

### 9.2. Improvement of Emulsification Technology for HIPEs 

For the fabrication of protein-based HIPEs, as mentioned earlier, a high-energy emulsification method is required. Few emulsification methods are available due to the high viscosity of HIPEs. Most of the reported protein-based HIPEs were fabricated based on high-speed homogenization. There are obvious disadvantages to this method. First, the emulsification mechanism of shear homogenization determines the bottom limit of the emulsion size, which is only suitable for the production of coarse emulsions. Compared to high-pressure homogenization, which only requires precise control of the pressure, flow rate and number of treatments, in practice, the emulsification effect of shear homogenization is susceptible to various influences. The size of the roto-stator, the depth of the rotor and stator into the liquid surface, the liquid volume, the direction of shear, etc., will all lead to this result, thus demanding production repeatability of the operator. It is necessary to introduce novel emulsification methods to prepare protein-based HIPEs with more uniform particle size distribution while providing convenient and precise control. 

### 9.3. Promoting the Research of HIPE Application in Foods 

For applications of protein-based HIPEs, it is crucial to ensure that the HIPEs are structurally stable and function correctly under realistic, complex conditions. For example, as an alternative to mayonnaise, fundamental studies only focus on consistent rheological properties of protein-based HIPEs with mayonnaise. Furthermore, researchers have designed protein-based HIPEs containing sucrose, NaCl, and acetic acid, which are closer to the actual conditions. Moreover, as an alternative to mayonnaise, HIPEs need to be considered in terms of odor, taste, nutritional value, etc., which is an elaborate task. Similarly, as a delivery system, it is not adequate to accomplish a single substance encapsulated in a simple system of HIPEs comprising water and oil under simulated experiments. 

### 9.4. Quality Control of Stabilizers and Protein-Based HIPEs

As mentioned above, the final formation of HIPEs from proteins is a multi-step process. Many factors affect the structure and properties of HIPEs, including stabilizers, oil phase, preparation conditions, and other parameters. The current studies only reveal the trends of factors on the structural properties of HIPEs in the presence of significant differences. Still, they do not point out the detailed standards that have distinguished different batches of HIPEs. Whether they can divide it using structural parameters (e.g., particle size, zeta-potential, interfacial film structure characteristics) or property parameters (stability, rheological properties, etc.), quality control of protein-based HIPEs is never mentioned in the reports.

## 10. Conclusions

The basic preparation of protein-based HIPEs includes protein collection, Pickering stabilizer fabrication, and emulsion design with emulsification. Due to the diversity of methods used to fabricate protein-based Pickering stabilizers, based on both covalent and non-covalent interactions, a wide range of protein types and sources are available. The protein sources have expanded from plants to animals and even to fungi, while the search for low-cost proteins and exploitation of neglected protein resources is a future trend. Some protein-based Pickering stabilizers are susceptible to environmental influences, and current attempts to use the protein complex particles. The potential of HIPEs in the food sector is undoubtedly huge, whether it is through the production of margarine, the delivery of nutrients, acting as a detergent, or preparing porous materials, HIPEs are a relatively new option. In the selection of HIPEs as a delivery system, the bioavailability of various bioactive substances can be effectively improved. Precise modulation of structural parameters through protein complex modifications is necessary to obtain stable, high-performance Pickering stabilizers in the research process. Relatively, monotonic emulsification technology limits the development of protein-based HIPEs, which is urgent to overcome. The application potential of protein-based HIPEs is unquestionable. However, current research mostly remains at the basic theoretical stage. Lastly, quality control and assurance of protein-based HIPEs were never considered.

## Figures and Tables

**Figure 1 foods-12-00482-f001:**
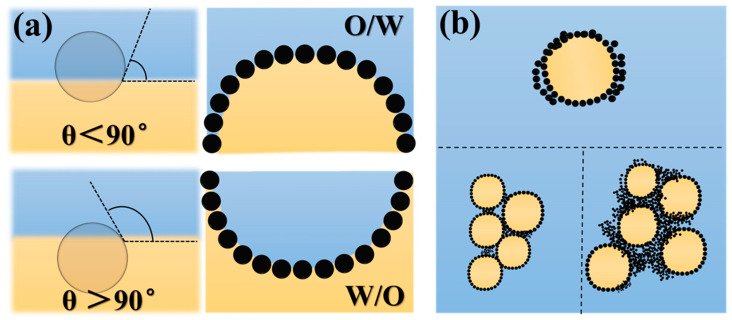
The oil phase (noted O) is in yellow, that the water phase (noted W) is in blue, and the black particles correspond to solid particles adsorbed at the interface. (**a**) Correlation of solid particle contact angle and Pickering emulsion type. (**b**) Structural characteristics of Pickering emulsion.

**Figure 2 foods-12-00482-f002:**
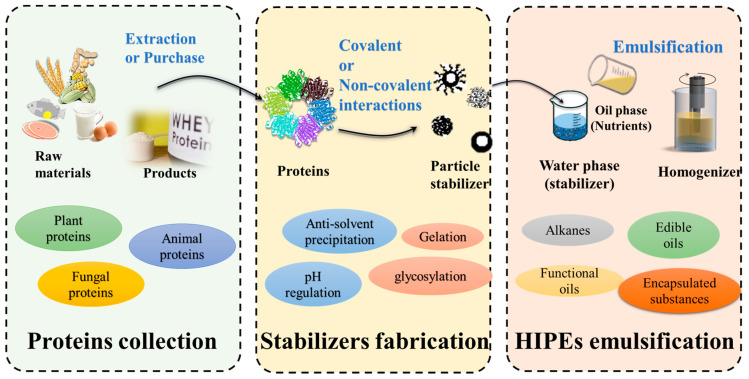
Preparation route of protein-based high-internal-phase Pickering emulsions (HIPEs).

**Figure 3 foods-12-00482-f003:**
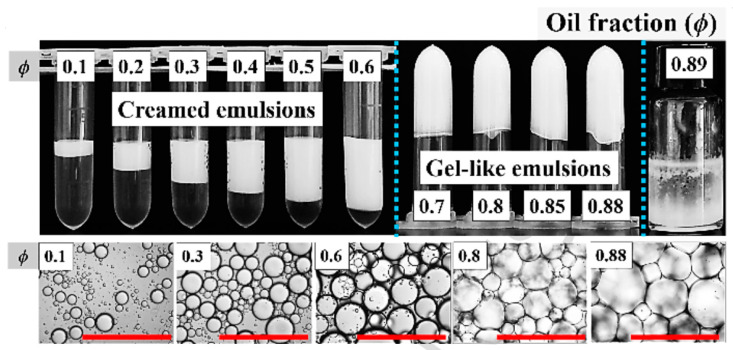
Apparent and microscopic characteristics of Pickering emulsions prepared with different volume fractions of oil phase. Adapted from [40] with permission from Elsevier Ltd., 2023.

**Figure 4 foods-12-00482-f004:**
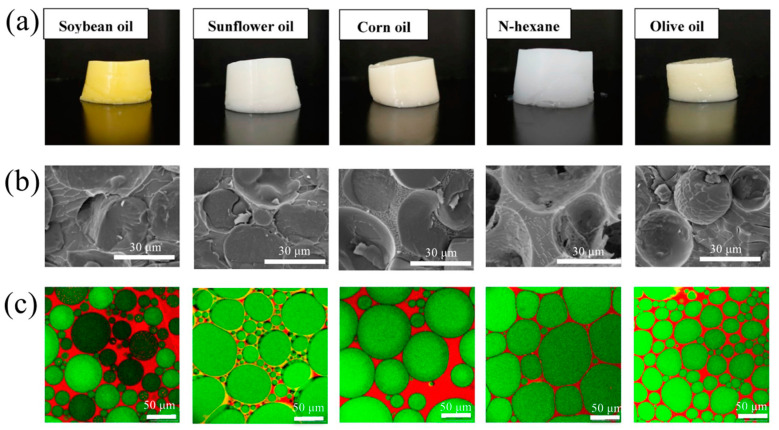
Characterization of the appearance of droplets of Pickering emulsions constructed with different oil phases: (**a**) appearance; (**b**) field emission electron microscopy; (**c**) CLSM. Adapted from [52] with permission from Elsevier Ltd., 2023.

**Figure 5 foods-12-00482-f005:**
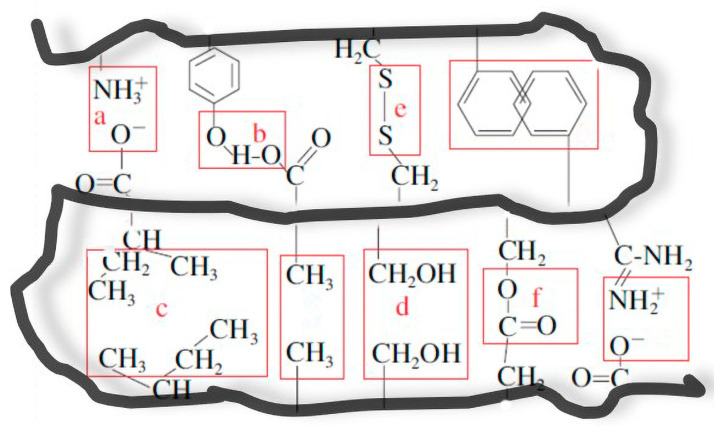
Protein molecular structure and chemical bonds: (**a**) ionic bonds; (**b**) hydrogen bonds; (**c**) hydrophobic bonds; (**d**) van der Waals forces; (**e**) disulfide bonds; (**f**) ester bonds.

**Figure 6 foods-12-00482-f006:**
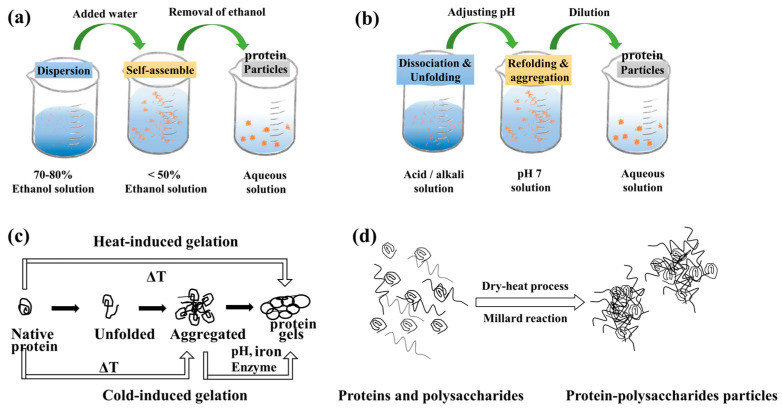
Main approaches and mechanisms for protein stabilizer preparation: (**a**) alcohol-soluble protein pretreatment; (**b**) preparation of HIPE stabilizers by pH adjustment method; (**c**) gelation process; (**d**) binding of proteins and polysaccharides by Maillard reaction.

**Figure 7 foods-12-00482-f007:**
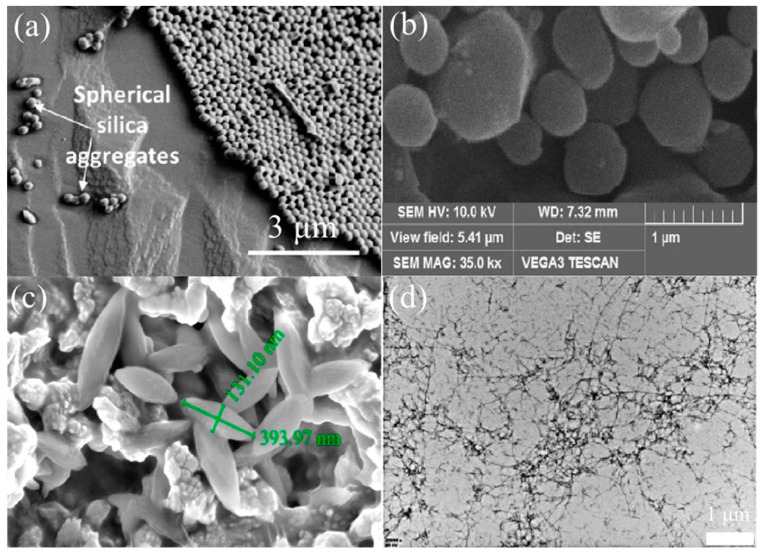
(**a**) Arrangement of spherical particles. (**b**) Microstructure of spherical particles. (**c**) Microstructure of spindle-shaped particles. (**d**) Microstructure of flocculent particles. Adapted from [42,85,98,103] with permission from Elsevier Ltd., 2023.

**Figure 8 foods-12-00482-f008:**
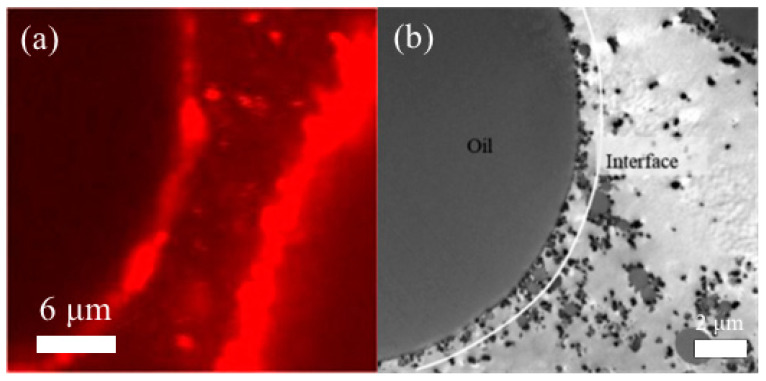
Particle disposition at the interface: (**a**) STED image of zein alcoholic protein stablized HIPE; (**b**) TEM image of droplet-stabilized emulsions. Adapted from [59,110] with permission from Elsevier Ltd., 2023.

**Table 1 foods-12-00482-t001:** Sources, composition, fabrication and characteristics of protein stabilizers.

Protein	Stabilizers	Preparation	Size (nm)	Contact Angle (°)	References
Zein	Zein–tannic acid complex particles	Anti-solvent	68–108	—	[46]
	Zein–pectin complex particles	Anti-solvent	583.74	86.44	[77]
	Zein–pga–rhamnolipid complex particles	Anti-solvent	504	84.05	[78]
	Zein protein particles–lecithin	Anti-solvent	100	—	[79]
Gliadin	Gliadin–chitosan complex particles	Anti-solvent	190–564	—	[80]
	Gliadin–chitosan complex particles	Anti-solvent	588.8	84.00	[26]
	Rice bran-modified wheat gluten nanoparticle	pH adjustment	250	104.6	[48]
Soy protein	Soy β-conglycinin–polyphenol complex nanoparticles	Anti-solvent	25.5–62.2	—	[81]
	Soy protein isolate–bacterial cellulose nanofibers complex particles	Anti-solvent	947–1177	—	[82]
	Soy protein microgels	Heat treatment	169.7–311.9	—	[31]
	Soy protein–polysaccharide complex particles	Ultrasonication	160	—	[17]
	Aggregation of soy β-conglycinin	Anti-solvent precipitation	500	—	[83]
	Globulin particles	pH adjustment	500–1000	—	[10]
Pea protein	Pea protein–high methoxyl pectin complex particles	pH adjustment	347	—	[84]
	Glycated pea protein isolate particles	Heat treatment	115.7–157.5	54.85–79.62	[39]
Peanut protein	Peanut protein microgel particles	Enzyme cross-link	200–300	102.3	[33]
Rice proteins	Rice protein–cellulose complex particles	pH adjustment	132–144	96.3–129.27	[29]
Fungal protein	Bamboo fungus protein gel particles	Enzyme cross-link	227	77.6	[35]
	Shiitake mushroom protein–polysaccharide conjugates	Direct extraction	300	75.5–115	[27]
Whey protein	Whey protein–low methoxyl pectin complex particles	pH adjustment	916–1032	60.45–70.15	[18]
	Whey protein microgels particles	Ca^2+^ induced	146.9	67	[30]
	Whey protein microgels	Heat treatment	90–350	—	[68]
	Whey protein nanofibrils	Hydrothermal method	200	—	[85]
BSA	BSA–trehalose complex particles	pH adjustment	5.3–8	—	[86]
	BSA–sucrose complex particles	pH adjustment	4–13	—	[87]
	Glycated BSA particles	Galactose glycated	4.2–6.5	—	[71]
Ovalbumin	S-ovalbumin particles	pH adjustment	5.8	—	[88]
	Ovalbumin–tannic acid complexes particles	pH adjustment	—	79.9–88.5	[89]
	Ovalbumin particles	pH adjustment	5.15	—	[90]
	Ovalbumin–pectin complex particles	pH adjustment	300–500	—	[91]
Casein	Casein nanogels particles	Glutaraldehyde cross-link	179	—	[92]
Meat protein	Pork proteins particles	pH adjustment	821–940	—	[34]
	Cod myofibers	pH adjustment	25–120	—	[93]
	Sea bass protein microgel particles	Enzyme cross-link	300–500	78.3–95.5	[94]
Gelatin type B	Gelatin particles	Glutaraldehyde cross-link	200	—	[95]
	Gelatin particles	Glutaraldehyde cross-link	235.9	—	[73]

The “—” indicates that this item has not been tested or published. The acronym BSA refers to bovine serum albumin.

**Table 2 foods-12-00482-t002:** Composition and functional properties of HIPE.

Oil Type	Encapsulated Compound	Stabilizer	Phase Fraction (%)	Droplet Size (μm)	Properties/Applications	References
Hexane	—	Gelatin particles	80	40	Degradable porous protein scaffold	[73]
Dodecane	β-carotene	Soy β-conglycinin particles	88	24–60	Heat, storage and freeze–thaw stability	[40]
	β-carotene	Ovalbumin particles	91	20–50	Heat, storage stability	[90]
	—	BSA–trehalose complex particles	80	20	Storage, heat stability	[86]
	—	Glycated BSA particles	92	30–50	Heat, storage and freeze–thaw stability	[71]
	—	Ovalbumin–tannic acid complexes particles	80	13–15	Storage and freeze–thaw stability	[89]
Squalane	—	Zein protein particles–soybean lecithin	80	25	Digestion stability	[79]
Corn oil	β-carotene	Gliadin–gum Arabic complex particles	85	6.0–9.4	Stability; nutrient protection	[37]
	β-carotene	S-ovalbumin particles	85	35	pH, ionic and temperature stability	[88]
	-carotene	Ca^2+^ induced WPI particles	80	16.6	Stability; enhance bioavailability	[38]
	Curcumin	Gliadin–chitosan complex particles	83	25.5–93.5	Nutrient protection; alternative for PHOs	[26]
	Curcumin	Zein–pectin complex particles	80	105.12	Stability; nutrient protection	[77]
	—	Soluble starch–whey protein isolate complex particles	75	5.55–10.60	pH, thermal and ionic stability	[138]
	Astaxanthin	Sea bass protein microgel particles	88	24–65	Enhance bioavailability; 3D printing	[94]
MCT oil	Curcumin	Whey protein nanogel particles	80	15	Enhance bioavailability and cellular uptake	[44]
	Lutein	Lysozyme–dihydromyricetin–mixture	90	48.5	Nutrient protection	[130]
	—	Zein–PGA–rhamnolipid complex particles	75	40–13	Environmental stability	[78]
	Indomethacin	Casein nanogels	80	50–160	Stability and tunable drug release	[92]
Olive oil	Quercetin	Pecan protein–xanthan gum complex particles	80	30–70	Enhance stability and bioavailability	[122]
	—	Fibrous protein particles	87	1–10	Heat, storage and freeze–thaw stability	[34]
Soy oil	Curcumin	Low density lipoprotein particles	80	10	Enhance stability and bioavailability	[131]
	—	Soy protein–polysaccharide complex particles	80	100	Storage, heat and freeze–thaw stability	[17]
	—	Globulin particles	80	562	Oxidation stability; nutrients delivery	[10]
	—	Aggregation soy β-conglycinin	80	25	Structural stability	[83]
	—	Rice protein–cellulose complex particles	85	20–60	3D printing	[29]
	—	Whey protein isolate–low methoxyl pectin	80	1400	Heating and centrifugal stability	[18]
Sunflower oil	β-carotene	Gelatin particles	80	10	Tunable structure and release behavior	[95]
	—	Zein–tannic acid complex particles	87	5.5–9.5	Microstructure tunable	[46]
	—	Soy Protein Microgels	82	8–12	Delayed digestive	[31]
	—	Cellulose nanofibers–soy protein complex particles	75	947–1409	Storage stability	[139]
Algal oil	Curcumin	Gliadin–chitosan complex particles	75	40	Protect and increase the bioavailability	[80]
Peanut oil	—	Peanut protein particles	87	10–30	Novel porous material template	[33]
	—	Whey protein nanofibrils	80	11	Stability; tunable rheology	[85]
	—	Shiitake mushroom protein–polysaccharide conjugates	78	72.3	Heat, storage stability; 3D printing	[27]
	—	Quinoa protein particles	80	14–24	Physical stability	[28]
Flaxseed oil	β-carotene	Soy β-conglycinin–polyphenol complex nanoparticles	80	50	Heat, storage stability; oxidation protection	[81]
Camellia oil	Cinnamaldehyde	Pea protein–pectin–EGCG complexes	83	—	Enhance stability; 3D printing	[140]

The “—” indicates that this item does not exist in the literature. The acronym MCT refers to medium-chain triglycerides. The acronym BSA refers to bovine serum albumin. The acronym PHO refers to propane hash oil. The acronym PGA refers to phosphoglyceric acid. The acronym EGCG refers to epigallocatechin gallate.

## Data Availability

Data is contained within the article.

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
