# Peer review of "Protein-Based High Internal Phase Pickering Emulsions: A Review of Their Fabrication, Composition and Future Perspectives in the Food Industry"

_foods, 2023, doi:10.3390/foods12030482_

Round 1

Reviewer 1 Report

This review under the title " Protein-based high internal phase Pickering emulsions: a review of their fabrication, composition and future perspectives in food industry " talked more about the components of the emulsion, even though the title mentions its application in the field of food, it is very short in two paragraphs

1-      I suggest to give more explanation and expansion in this field “Application in the food field like encapsulation, replacing….”

2-      I suggest that explain about the classification like microsphere, microcapsule, colloidal products ….

3-      I suggest that explain about Enteral factors affecting Pickering emulsion stability

4-      Rewrite the introduction and conclusion with more emphasis on application in the food industry

Author Response

Reviewer 1

This review under the title " Protein-based high internal phase Pickering emulsions: a review of their fabrication, composition and future perspectives in food industry " talked more about the components of the emulsion, even though the title mentions its application in the field of food, it is very short in two paragraphs

  • I suggest to give more explanation and expansion in this field “Application in the food field like encapsulation, replacing….”

As suggested, the information was added in lines 594-698.

  • I suggest that explain about the classification like microsphere, microcapsule, colloidal products ….

As suggested, the information was added in lines 669-698.

  • I suggest that explain about Enteral factors affecting Pickering emulsion stability

As suggested, the enteral factors were added in lines 67-77.

  • Rewrite the introduction and conclusion with more emphasis on application in the food industry.

The introduction and conclusions have been partially revised. The information was provided in lines 766-770. Thanks.

Reviewer 2 Report

This is an interesting review paper about protein-based HIPEs.

However, this paper could be improved. For example, the authors should show microscopy images of HIPEs (visualization of the droplets and of the particles at the interface). Also, the authors should discuss the physical stability or unstability of HIPEs. The size of the particles (nanoparticles ?) should be further discussed, as well as their morphology (in aqueous media, and after adsorption at interfaces).

The references cited could be improved.

Details comments:

1./Abstract :

The vocabulary « droplet » should appear in the abstract.

Line 19 – 20 : Stabilizers are irreversibly adsorbed at the (dispersed phase – continuous phase) interface or at the surface of droplets to maintain the structure of the droplets.

Line 26 : oil phase or internal phase ?

2./Text :

Line 39 : after the introduction of « emulsion » (line 34-39), the authors should define the « interface » and its key role in the physical stabilization of the droplets in emulsions.

Line 42 : add the reference (Pickering, 1907) [3].

Line 74-75 : the sentence « For HIPEs » is not clear.

Line 109 : please explain the impact of the solid particle size for the stabilization of HIPEs. Please explain why nanoparticles are used. The term « particles » is also used (fig. 4) ; could the authors clarify the importance of the size of the particles.

Line 125 – 127 : please clarify this sentence.

Line 130-131 : the authors should introduce the notion of « interface » between the oil and the water phases ; and the notion of « Pickering particles » as stabilizers.

Line 132 : by « particle », the authors refer to « stabilizers » ?

Line 162 : please change « stabilizers » by « protein-based stabilizers »

Line 181 : please remove the vocabulary « particle stabilizer » ; at the stage of preparation, « protein-based particles » are obtained ; their role as stabilizers is not confirmed. Same comment line 183.

Line 184 – 185 : the authors used the vocabulary « particles » and « nanoparticles » ; once again, if the size of the particles is important, this shoud be detailed in the review paper.

Line 402 : please correct « HIPPE »

Line 412 : the authors introduce the vocabulary « emulsifiers », and should discuss the difference with « stabilizers »

After line 417 ….. the lines were not numbered ….. which increases the difficulty for the reviewer.

Paragraph 5.1 dedicated to the oil phase should be improved. Concerning the oil phase, the authors should mention the melting point (liquid oil, dolid fat)

The paragraph from « N-hexane » is not clear … The authors should detail the hydrocarbons that are used to prepare HIPEs.

3./Figures :

Figure 1 : This figure could be improved by adding some more information. Please indicate that the oil phase (noted O) is in yellow, that the water phase (noted W) is in blue, and that the black particles correspond to solid particles adsorbed at the interface. In figure (a) the terme interface should appear. In the text, the method(s) used to determine the contact angle should be precised.

Figure 2 : the abreviation « HIPEs » should be defined.

Figure 3 : this figure contains letters « a, b, c, d, e, f » that should be defined.

Figure 4 : this figure contains 4 parts noted (a), (b), (c), (d) that should be defined ; add sub-titles. The word « protein » should be introduced in fig (a), (b) and (c).

 4./Tables

Table 1: please define the abreviation "BSA"

Table 2 : please define the abreviations such as MCT, BSA, PHOs, PGA, EGCG …

Author Response

Reviewer 2

This is an interesting review paper about protein-based HIPEs.

However, this paper could be improved. For example, the authors should show microscopy images of HIPEs (visualization of the droplets and of the particles at the interface). Also, the authors should discuss the physical stability or unstability of HIPEs. The size of the particles (nanoparticles?) should be further discussed, as well as their morphology (in aqueous media, and after adsorption at interfaces).

The references cited could be improved.

References have been increased appropriately, thanks.

Details comments:

1./Abstract :

The vocabulary « droplet » should appear in the abstract.

As suggested, “droplet” was added in lines 13.

Line 19 – 20: Stabilizers are irreversibly adsorbed at the (dispersed phase – continuous phase) interface or at the surface of droplets to maintain the structure of the droplets.

The original expression has been modified according to the suggestion (line 14). Thanks.

Line 26: oil phase or internal phase ?

Oil phase, here we want to express the system of HIPEs consisting of stabilizer, water phase, oil phase and loading substance (line 26).

2./Text :

Line 39 : after the introduction of « emulsion » (line 34-39), the authors should define the « interface » and its key role in the physical stabilization of the droplets in emulsions.

Discussion about the interface has been added (lines 135-171).

Line 42 : add the reference (Pickering, 1907) [3].

As suggested, the reference was cited (lines 34).

Line 74-75 : the sentence « For HIPEs » is not clear.

Here wanted to express that there is also variability in the stability of HIPEs constructed with the participation of different matrix particles. The phrases were written (lines 60-62).

Line 109 : please explain the impact of the solid particle size for the stabilization of HIPEs. Please explain why nanoparticles are used. The term « particles » is also used (fig. 4) ; could the authors clarify the importance of the size of the particles.

Solid particle size has an effect on the stability of HIPEs but the trend is not unique. The stability of HIPEs is also affected by the wettability of the particles, or by the acidic and alkaline environment. The focus here is on the protein matrix as a stabilizer, and nanoparticles are one of the classifications of protein matrix particles, The discussion on particle size has been added (lines 291-311).

Line 125 – 127: please clarify this sentence.

Since the oil and water phases do not naturally dissolve each other, high speed homogenization is a common method for mixing the oil and water phases. The original expression has been modified (lines 122-123).

Line 130-131: the authors should introduce the notion of « interface » between the oil and the water phases ; and the notion of « Pickering particles » as stabilizers.

Discussion about the interface has been added (lines 135-171).

These concepts have been more specifically described (lines 148-138).

Line 132 : by « particle », the authors refer to « stabilizers » ?

The particles here represent the droplets under the microscope after the HIPEs have been constructed (lines 122-123).

Line 162 : please change « stabilizers » by « protein-based stabilizers »

The original expression has been modified (line 190).

Line 181 : please remove the vocabulary « particle stabilizer » ; at the stage of preparation, « protein-based particles » are obtained ; their role as stabilizers is not confirmed. Same comment line 183.

The original expression has been modified (line 204).

Line 184 – 185 : the authors used the vocabulary « particles » and « nanoparticles » ; once again, if the size of the particles is important, this shoud be detailed in the review paper.

The size of the particles has been specified more specifically (line 264-311).

Line 402 : please correct « HIPPE »

The original expression has been modified (line 496).

Line 412 : the authors introduce the vocabulary « emulsifiers », and should discuss the difference with « stabilizers »

The emulsifier here is a stabilizer, and in the field of emulsions, the emulsifier is a type of stabilizer.

After line 417 ….. the lines were not numbered ….. which increases the difficulty for the reviewer.

The lines were numbered.

Paragraph 5.1 dedicated to the oil phase should be improved. Concerning the oil phase, the authors should mention the melting point (liquid oil, dolid fat)

The discussion on melting point has been added (line 550-563).

The paragraph from « N-hexane » is not clear … The authors should detail the hydrocarbons that are used to prepare HIPEs.

 The original expression has been modified (line 524-535).

3./Figures:

Figure 1 : This figure could be improved by adding some more information. Please indicate that the oil phase (noted O) is in yellow, that the water phase (noted W) is in blue, and that the black particles correspond to solid particles adsorbed at the interface. In figure (a) the terme interface should appear. In the text, the method(s) used to determine the contact angle should be precised.

The original expression has been modified according to the suggestions (line 783-785). Thanks.

Figure 2: the abreviation « HIPEs » should be defined.

The original expression has been modified (line 787).

Figure 3: this figure contains letters « a, b, c, d, e, f » that should be defined.

Due to modifications, Figure 3 became Figure 5. The original expression has been modified (line 795-797)

Figure 4: this figure contains 4 parts noted (a), (b), (c), (d) that should be defined ; add sub-titles. The word « protein » should be introduced in fig (a), (b) and (c).

As suggested, the 4 parts including (a), (b), (c), (d) were defined. The subtitle has been added (line 799-801).

Due to modifications, Figure 4 became Figure 6 and has been supplemented with protein in Figure 6.

4./Tables

Table 1: please define the abbreviation "BSA"

As suggested, the abbreviation “BSA” was defined (line 808).

Table 2: please define the abbreviations such as MCT, BSA, PHOs, PGA, EGCG …

As suggested, the abbreviations were defined (line 812-814).

Round 2

Reviewer 1 Report

The author has responded and implemented the reviewer's comments well and has done a good rewriting.